# Trusted re-weighting for label distribution learning

**Zhuoran Zheng**[1]      **Chen Wu**[2]      **Yeying Jin**[3]      **Xiuyi Jia**[*1]

[1]School of Computer Science and Engineering, Nanjing University of Science and Technology, Nanjing, China
[2]University of Science and Technology of China, Hefei, China
[3]Department of Electrical and Computer Engineering, National University of Singapore, Singapore

## Abstract

Label distribution learning (LDL) is a novel machine learning paradigm that aims to shift 0/1 labels into descriptive degrees to characterize the polysemy of instances. Since the description degree takes a value between 0∼1, it is difficult for the annotator to accurately annotate each label. Therefore, the predictive ability of numerous LDL algorithms may be degraded by the presence of noise in the label space. To address this problem, we propose a novel stability-trust LDL framework that aims to reconstruct the feature space of an arbitrary LDL dataset by using feature decoupling and prototype guidance. Specifically, first, we use prototype learning to select reliable cluster centers (representative vectors of label distributions) to filter out a set of clean samples (with labeled noise) on the original dataset. Then, we decouple the feature space (eliminating correlations among features) by modeling a weight assigner that is learned on this clean sample set, thus assigning weights to each sample of the original dataset. Finally, all existing LDL algorithms can be trained on this new re-weighted dataset for the goal of robust modeling. In addition, we create a new image dataset to support the training and testing of compared models. Experimental results demonstrate that the proposed framework boosts the performance of the LDL algorithm on datasets with label noise.

## 1 INTRODUCTION

Currently, label distribution learning (LDL) plays a landmark role in characterizing task uncertainty and conveying the polysemy of an instance [Gao et al., 2017, Geng,

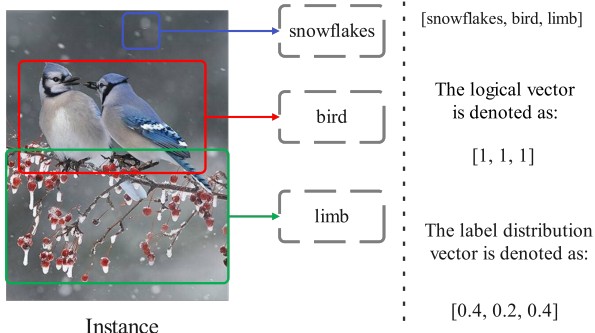

Figure 1: This figure visualizes the difference between label distribution learning and multi-label learning describing a single instance. If the description object is an image, the label distribution value represents the percentage of components in the image.

2016, Zheng et al., 2021, Zheng and Jia, 2022]. In contrast to the classical multi-label learning paradigm [Zhang and Zhou, 2013], LDL describes an instance as a distribution of descriptive degrees rather than a vector of 0/1 labels (see Figure 1). Therefore, a learner (classifier or regressor) tends to focus more on tracking the decision bounds, hence the robustness of the whole algorithm is boosted [Le et al., 2023].

Recently, a large body of work [Chen et al., 2021, Gao et al., 2018, Li et al., 2022d, Liu et al., 2021, Si et al., 2022, Zhao et al., 2021] leverages the properties of LDL to characterize the relation between feature space and label space for achieving competitive performance. However, most researchers overlook the fact that the label space of the dataset may be noisy since the uncertainty of manual annotation and the inductive bias of the label enhancement algorithm [Xu et al., 2019] can introduce noise into the label space (for example, the annotators misrecorded the percentage of the two components and exchanged their label distribution values). This low-quality set of labels can cause the LDL algorithm to be off the right modeling track, usually showing up as under-performance on the test set.

*Correspondence. This work was supported by the National Natural Science Foundation of China (62176123).

*Accepted for the 40th Conference on Uncertainty in Artificial Intelligence* (UAI 2024).

Several works [Li et al., 2022c, Zheng and Jia, 2022] attempt to address the problem of label noise on LDL benchmarks. Li et al. [2022c] by building expert knowledge to endow the training set samples with different weights during the model iteration. Zheng and Jia [2022] estimate label uncertainty by building a label distribution matrix on the label space. Existing work on noise processing is embedded in the prediction algorithm, which is tightly coupled with the algorithm and has low generalizability. In contrast, we develop a generalized LDL pre-processing framework (stability-trust framework) in this paper, which is a catch-all paradigm for constructing high-quality LDL datasets to serve existing LDL algorithms to boost performance. Specifically, first, we attempt to create a pseudo-label space over the label distribution space, to distinguish which samples may be noisy with the help of the prototype guidance. Then, a clean training set is filtered by a look-up table for training an efficient weight assignor (when the weight assignor is trained on a noisy dataset it leads to a degradation of the algorithm performance). Note that the prototype space includes several vectors that can represent the characteristics of the label distribution space. For instance, if the label distribution space consists of six label values, we divide the dataset into six subsets and estimate their expectations to obtain six representative vectors. Finally, a weight assignor reconstructs the original LDL dataset using a feature decoupling scheme (including a kernel map). Feature decoupling is to treat the feature space of the customized LDL datasets as tabular information and decouple the correlation between features by assigning weights to the samples. However, this weight assigner predicts that the label distribution can be overly compact due to the prototype guide, and for this reason, we consider using uncertainty modeling to assign tiny weights to a small number of samples as a positive incentive for noise [Li, 2022]. The contributions of this paper are summarized as:

**1)** We propose a stability-trust framework that achieves the dual purpose of denoising and feature decoupling by assigning weights to the raw sample space that helps the downstream learners enhance their regression abilities.

**2)** We use a prototype space to guide the weight assignor to inscribe a compact learning space for faster convergence of the model. In addition, we consider an uncertainty modeling algorithm to construct some positive incentive noise to boost the performance of the learner.

**3)** In contrast to the existing tabular datasets (customized LDL dataset), we build a new image dataset stored in the form of image-to-label[1] to evaluate the deep networks.

---

[1]https://github.com/zzr-idam/LDL

## 2 RELATED WORK

**Label distribution learning.** Currently, LDL plays a vital role in estimating a task's uncertainty and thus boosting the model generalization capability. LDL is similar to deep learning modeling approaches, where the output of a model is usually standardized into probability vectors by Softmax. However, in contrast, LDL gives semantic information and a priori distributional constraints, which can allow it to be used as a regularization term to help improve the performance of existing methods. The LDL paradigm is built from an age estimation task [Geng, 2016]. Since then a large number of approaches have been proposed, such as low-rank hypothesis-based [Jia et al., 2019, Ren et al., 2019b], metric-based [Gao et al., 2018], manifold-based [Tan et al., 2022, Wang and Geng, 2021], and label correlation-based [Qian et al., 2022, Teng and Jia, 2021]. Moreover, some approaches are implemented in computer vision [Chen et al., 2021, Gao et al., 2018, Li et al., 2022a, Zhao et al., 2021], and speech recognition [Si et al., 2022] tasks to boost the performance of classifiers. Recently, several approaches based on LDL start to tackle the label noise problem [Li et al., 2022c, Zheng and Jia, 2022]. However, these approaches are customized strategies, and we attempt to build a generalized preprocessing method to serve extant LDL algorithms.

**Prototype learning.** Prototype learning [Deng et al., 2021, Dong and Xing, 2018, Li et al., 2021, Ren et al., 2022, Wang et al., 2021, Yang et al., 2018] is a classical learning paradigm in machine learning and pattern recognition, which aims to select a representative subset to guide the behavior of downstream tasks. For example, the nearest neighbor algorithm (KNN) [Guo et al., 2003] is a typical prototype learning case, which guides the aggregation of the whole dataset by obtaining the centroids of a cluster. Currently, prototype learning is utilized in several domains, such as image recognition, speech recognition [Rouat and Garcia, 2021], and inference of textual content [Haghighi and Klein, 2006]. In the LDL domain, prototype learning plays the role of feature selection to help downstream LDL learners [González et al., 2020]. In this paper, the prototype learning paradigm is used to help model a weight assignor by filtering a clean subset.

**Label noise estimation.** The existing label space of large datasets hardly avoids the disturbance of noise, due to the complexity of the task, the subjectivity of the annotator, the inaccuracy of the annotation algorithm, etc. Based on this, numerous works are presented to address the problem of noise disturbance [Arazo et al., 2019, Ju et al., 2022, Kaneko et al., 2019, Li et al., 2022b, Reeve and Kabán, 2019, Xie and Huang, 2022, Yao et al., 2020, Zhu et al., 2021]. There are two main strategies to solve such problems, one is to build a robust learning target or regularization term, and the other is to renovate the model for unbiased estimation. In the

field of LDL, there are already some works [Li et al., 2022c, Zheng and Jia, 2022] that consider the presence of noise in the label space, however, these works are only applicable to customized LDL algorithms. In contrast, our algorithm is a general framework as a data pre-processing technique.

## 3 STABILITY-TRUST FRAMEWORK

In this paper, we develop a stability-trust framework focusing on tackling the problem of label distribution datasets with noisy labels. Be aware that our framework can also handle multi-class tasks with noisy labels.

**Notation.** Given a particular instance, the goal of LDL is to learn the degree to which each label describes that instance. Input matrix (tabular data) $\mathbf{X} \in \mathbb{R}^{M \times N}$, where $M$ is the number of instances and $N$ is the dimension of features. $x_i$ is the $i$-th instance in the dataset. The label distribution space is defined as $\mathcal{D} \in \mathbb{R}^{M \times L}$, and $\mathcal{D}_j$ is the $j$-th label. For each instance $x_i$, its label distribution is $\mathcal{D}_i = \left\{ d_{x_i}^{y_1}, d_{x_i}^{y_2}, \cdot, d_{x_i}^{y_L} \right\}$, where $d_{x_i}^{y_j}$ is the description degree of the label $y_j$ for $x_i$. The $d_{x_i}^{y_j}$ is constrained by $d_{x_i}^{y_j} \in [0, 1]$ and $\sum_{j=1}^{L} d_{x_i}^{y_j} = 1$. In addition, the prototype space is defined as $\mathcal{P} \in \mathbb{R}^{L \times L}$, then the prototype vector is defined as $p_j$. The virtual label vector of all instances guided by the prototype learning is $\mathcal{VL} = \{vl_1, ..., vl_M\}$. The label distribution that is predicted by the model is defined as $\mathcal{L}_i = \left\{ l_{x_i}^{y_1}, l_{x_i}^{y_2}, ..., l_{x_i}^{y_L} \right\}$. Building a pseudo-label vector on the label space $\mathcal{Y}$ is $\mathcal{Q} = q_1, q_2, \cdot, q_M$ , and $q_i$ denotes the pseudo-label for instance $x_i$.

**Assumptions.** We rely on three key principles or assumptions for developing a stability-trust framework. **a)** Prototypes are usually the information least disturbed by noise, such as the output of the mean filter and adaptive weighted average filter. The prototype space as a "clean" set can push the predictive distribution of the model closer to the central data distribution. In other words, using the prototype space as a guiding principle may lead to the construction of a new sample space that is more compact within the class and expands the distance between classes. For blind datasets with noisy labels, this strategy yields a high-quality set with minimal outlay. **b)** We use the prototype space to check the estimated flags on the label distribution against the flags that are self-contained by the label distribution to filter out high-quality learning space for the weight assignor. This is an efficient filtering mechanism that uses the consistency of these two flags as the base for whether the sample is credible or not. **c)** Based on the stable learning paradigm [Shen et al., 2020], we attempt to improve the inference ability of the classifier by decoupling the correlation between features. Specifically, stable learning uses a tactic of assigning weights to samples to achieve feature decoupling, and the overall framework can be written as Algorithm 1. Here $w$ can be a linear algorithm or a deep network, and $\hat{\beta}$ works ultimately on the raw sample space $\mathbf{X}$.

---

**Algorithm 1** Stable Learning Framework

1: **Input** : Dataset $\mathcal{B} = \{\mathbf{x}^{(i)} = (x_1^{(i)}, ..., x_d^{(i)}), y^{(i)}\}_{i=1}^n$
2: **Output** : Coefficients $\hat{\beta}$ on each variables
3: /*Step I*/
4: Learn weight $w(\mathbf{X})$ to make $\mathbf{X}$ are mutually independent of each other.
5: /*Step II*/
6: Solve weighted least squares with weighting function $w(\mathbf{X})$. The solution is $\hat{\beta}_w^{(n)}$.
7: Return $\hat{\beta}_w^{(n)}$.

---

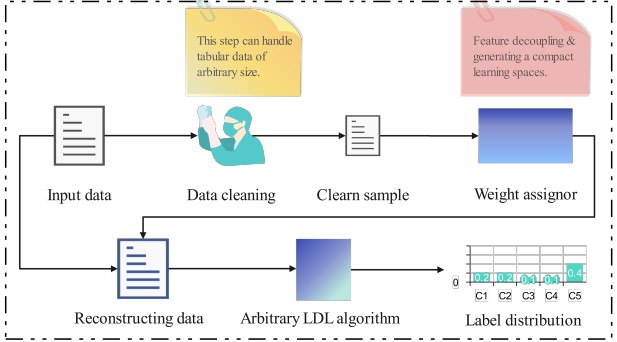

(a) Workflow of our algorithm.

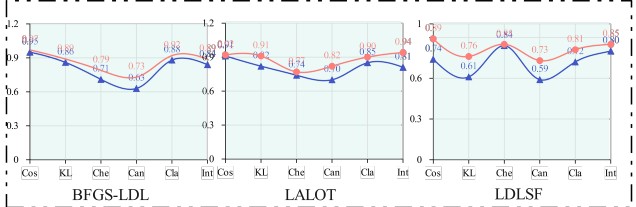

(b) Comparison of the performance of the methods on the fbp5500 dataset.

Figure 2: **Our architecture.** This figure(a) shows the architecture of the proposed stability-trust, which consists of three parts. This figure(b) represents the performance of the three LDL algorithms on the dataset, with the blue line indicating the training set without the modification and the red line indicating the dataset with reconstruction scheme. The data in Figure(b) are normalized to be between 0 and 1. Predictably, images that are integrated over a curvilinear surface show better performance with larger areas.

**Goal.** Although the framework aims to assign weights to each sample in the raw sample space formally, it has two key goals. On the one hand, decoupling the correlation between features constructs a stable and robust learning space. On the other hand, the output space of the network is guided by prototypes to create a high-quality training set with compact intra-class distance and relaxed inter-class distance. However, the principle of prototype learning overly makes the model's predictions compact, and to alleviate this problem, we add a moderate amount of noise to the training set.

# 4 PROPOSED METHOD

As shown in Figure 2, our framework is divided into three stages, first, we use a standard prototype learning to filter out a relatively clean training set $\hat{\mathcal{B}}$; next, we design some customized loss terms by which a corresponding coefficient is learned for each instance $x_i$ of the raw space; finally, potentially noisy instances are given slight weights as positive incentive noise as part of the training set. This approach is a plug-and-play data pre-processing strategy to model arbitrary sizes of tabular data sets.

**Obtaining a clean set $\hat{\mathcal{B}}$ from the raw space $\mathcal{B}$.** Faced with a label distribution dataset of arbitrary size with label noise $\mathcal{B}$, we need to simply clean it with the help of the prototype space $\mathcal{P}$. The purpose of cleaning dataset $\mathcal{B}$ to obtain $\hat{\mathcal{B}}$ is to provide a high-quality training set for generating a weight assignor. So far, one question needs to be discussed, *why do we require prototype learning to guide the reconstruction of datasets*? We visualize the label space of a label distribution dataset as shown in Figure 3.

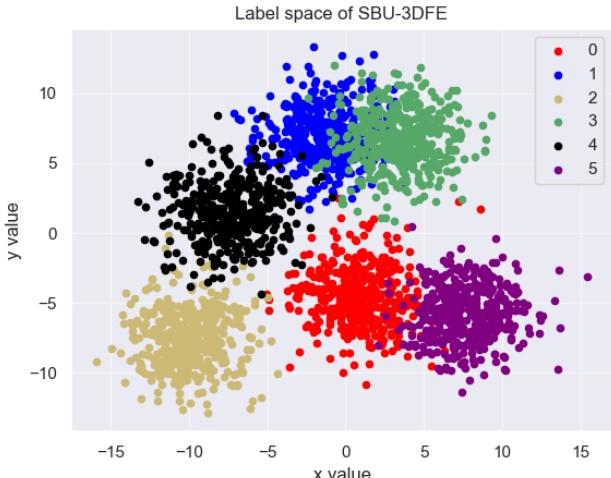

Figure 3: We visualize the label space of the SBU-3DFE dataset by using the t-SNE algorithm [Van Der Maaten, 2014], where t-SNE is based on the KPCA algorithm Anowar et al. [2021]. Intuitively, the label distribution space of SBU-3DFE can be viewed as having 6 clusters, a property that exactly matches the dimensionality of the label space. Even if the label distribution space is noisy, the center position of each cluster can still serve as reliable target information.

The label space of this dataset (SBU-3DFE) has 6 dimensions, which correspond exactly to the 6 clusters in Figure 3. We leverage prototype learning to sieve out representative vectors of each cluster as prototypes $p_j$. Specifically, we start with building the prototype space $\mathcal{P}$ on the training dataset $\mathcal{B}$. In the first step, $L$ subsets are constructed, and each subset stores the vectors $\mathcal{D}$ that can represent this label. In the second step, the mean values in each of the $L$ sets are obtained as a prototype to build a prototype space of size

$L \times L$. The formal expression under the Python style:

$$\text{prototype}[j, :] = \text{mean}(\mathcal{D}\underbrace{[\text{where}(\mathcal{D}_i[j] > (1/L)), :]}_{\text{prototype vector}}), j \in L. \tag{1}$$

Following the prototype space $\mathcal{P}$ being constructed, we introduce how to build the pseudo-label vector $\mathcal{Q}$ and the prototype learning guided virtual label vector $\mathcal{VL}$. In label distribution space $\mathcal{D}$, the index number of the maximum value in each label distribution $\mathcal{D}_i$ is assigned as the pseudo-label $q_i$ for instance $x_i$. For example, for the instance with label distribution [ 0.1, 0.1, 0.1, 0.4, 0.1, 0.2 ], its pseudo-label is 4. For the virtual label vector $\mathcal{VL}$ guided by prototype learning, we apply KNN (K=1) on the prototype space $\mathcal{P}$ to search the virtual label for each instance. For example, for $x_i$, we calculate the Euclidean distance to each prototype in the prototype space $\mathcal{P}$, to select the nearest prototype $p_j$ and use the index number with the maximum value in prototype $p_j$ as the virtual label of $x_i$. Finally, compare the constructed $\mathcal{Q}$ and $\mathcal{VL}$, if the paired $q_i$ and $vl_i$ take the same value, keep the instance $x_i$ and its corresponding label distribution $\mathcal{D}_i$ to obtain a new training set $\mathcal{B}$.

**Learning the coefficients of the raw samples.** We try to design a set of coefficients assigned $\hat{\beta}_w^{(n)}$ to the raw samples.

Specifically, we use a simple linear model to learn these coefficients in an end-to-end manner. First, we introduce the **stable learning** problems as follows:

*Problem.* Given the target $\hat{\beta}_w^{(i)}$ and input variables $x = [x_1, ...x_s] \in \mathbb{R}^s$, the task is to learn a predictive model which can achieve uniformly small error on any data point.

We consider the linear regression problem with model misspecification. Specifically, we can assume the target $\hat{\beta}_w^{(i)}$ is generated by the following form:

$$\hat{\beta}_w^{(i)} = x^\top \mathbf{W}_{1:s} + \mathbf{W}_0 + b(x) + \epsilon, \tag{2}$$

where $x \in \mathbb{R}^s$ is an input vector, $b(x)$ is a bias term that depends on $x$, such that $|b(x)| \geq \delta$ and $\epsilon$ is zero-mean noise with variance $\sigma^2$. Next, we need to use this model to build a set of training data with an optimization target to generate $\hat{\beta}_w^{(i)}$. Here, we eliminate the values of the non-diagonal elements (ND) of the correlation matrix with the help of an L2 norm.

$$\min ||\sum_{i=1}^{N} \text{ND}(((\hat{\mathbf{X}}^\top \mathbf{W})\mathbf{X})(\hat{\mathbf{X}}^\top \mathbf{W})\mathbf{X})^\top)_i - \rho||_2, \tag{3}$$

where $\hat{\mathbf{X}}$ denotes the feature space of a clean set of samples $\hat{\mathcal{B}}$, $\mathbf{X}$ denotes feature space of raw samples corresponding to $\mathcal{B}$, and $\rho$ denotes a small number ($\rho \leq 0.01$). Note that ND assembles the non-diagonal elements of a square matrix into a one-dimensional array. Predictably, we only address the correlations that exist between features in a linear space. High-order correlations may still exist, and

to completely decouple the correlations between features, we kernel-mapped (Gaussian kernel: $e^{-\frac{||x-x'||^2}{2\sigma^2}}$) the reconstruction matrix ($\ker(\hat{\mathbf{X}}^\top \mathbf{W})\mathbf{X})$). The optimization objective of this algorithm can be written:

$$\min ||\sum_{i=1}^{N} \text{ND}(((\hat{\mathbf{X}}^\top \mathbf{W})\mathbf{X})((\hat{\mathbf{X}}^\top \mathbf{W})\mathbf{X})^\top)_i - \rho||_2 +$$

$$\lambda||\sum_{i=1}^{N} \text{ND}(\ker(((\hat{\mathbf{X}}^\top \mathbf{W})\mathbf{X}))\ker((\hat{\mathbf{X}}^\top \mathbf{W})\mathbf{X})^\top))_i - \rho||_2, \quad (4)$$

where $\lambda$ denotes a hyperparameter, which is obtained by parameter sensitivity analysis. In addition, we add a regularization term for $\mathbf{W}$. The overall optimization objective can be written as follows:

$$\min ||\sum_{i=1}^{N} \text{ND}(\bar{\mathbf{X}}\bar{\mathbf{X}}^\top)_i - \rho||_2 + \lambda||\sum_{i=1}^{N} \text{ND}(\ker(\bar{\mathbf{X}})\ker(\bar{\mathbf{X}})^\top)_i$$

$$- \rho||_2 + \gamma||\mathbf{W}||_1, \quad (5)$$

where $\bar{\mathbf{X}} = \hat{\mathbf{X}}^\top \mathbf{W}\mathbf{X}$, $\gamma$ is a hyperparameter. To eliminate the higher-order correlation between features, we apply a soft trick whose values of the diagonal elements of $\text{tr}(\bar{\mathbf{X}}\bar{\mathbf{X}}^\top)$ tend to 1. The approach considered in this paper is motivated by the following theoretical result, which shows the effect of model misspecification bias even when the sample size is infinity.

*Proposition*. Consider the L2 norm when the sample size is infinity:

$$\hat{\beta} = \min \mathbf{E}_{(\mathbf{X},\rho)}(\sum_{i=1}^{N} \text{ND}(\bar{\mathbf{X}}\bar{\mathbf{X}}^\top)_i - \rho)^2. \quad (6)$$

The estimation bias caused by the worst case perturbation error $|b(x)| \leq \delta$ can be as bad as $||\hat{\beta} - \mathbf{W}||^2 \leq 2(\delta/\psi) + \delta$, where $\psi^2$ is the smallest eigenvalue of $\mathbf{E}((\mathbf{X} - \mathbf{E}(\mathbf{X}))(\mathbf{X} - \mathbf{E}(\mathbf{X}))^\top)$.

*Proof.* Let $\Delta\mathbf{W} = \mathbf{W} - \bar{\mathbf{W}}$ and $\Delta\hat{\beta} = \hat{\beta} - \bar{\mathbf{W}}$. We have $\Delta\hat{\beta} = \min \mathbf{E}(\mathbf{X}\Delta\mathbf{W} - b(\mathbf{X}))^2$. At the optimal solution, we have $\Delta\hat{\beta} = \mathbf{E}(b(\mathbf{X})) - \mathbf{E}(\mathbf{X}^\top \Delta\hat{\beta}_{1:s})$. By elimination $\mathbf{W}_0$, and let $\tilde{\mathbf{X}} = \mathbf{X} - \mathbf{E}(\mathbf{X})$, and $\tilde{b}(\mathbf{X}) = b(\mathbf{X}) - \mathbf{E}_{\mathbf{X}}(b(\mathbf{X}))$, we have $\Delta\hat{\beta}_{1:s} = \min(\tilde{\mathbf{X}}^\top \Delta\mathbf{W}_{1:s} - \tilde{b}(\mathbf{X}))^2$. It follows that $\Delta\hat{\beta}_{1:s} = (\mathbf{E}(\tilde{\mathbf{X}})\tilde{\mathbf{X}}^\top)^{-1}\mathbf{E}(\tilde{b}(\mathbf{X}))\tilde{\mathbf{X}}$. This implies that $\Delta\hat{\beta}_{1:s} \leq \delta/\psi$. Moreover, it implies that $|\Delta\hat{\beta}_0 \leq \delta + \delta/\psi|$. We thus obtain the desired bound.

In the proposition, we observe that the worst-case estimation error tends to infinity when $\psi$ tends to 0. This means that when the variables are highly co-linear, ordinary least squares yield a bad solution even when the training data is very large (or infinite). To solve this problem, we introduce the re-weighting theorem in [Shen et al., 2020] to alleviate this problem. This strategy leads to a total bias that is a

Table 1: Statistics of the experimental datasets. $\hat{\mathcal{B}}$ denotes a relatively clean dataset obtained from the raw sample space.

| ID | Dataset | Examples | Features | Labels | $\hat{\mathcal{B}}$ | Full-rank |
|----|---------|----------|----------|--------|---------|-----------|
| 1 | wc-LDL | 500 | 243 | 12 | 163 | Yes |
| 2 | JAFFE | 213 | 243 | 6 | 180 | Yes |
| 3 | SBU-3DFE | 2500 | 243 | 6 | 156 | Yes |
| 4 | Scene | 2000 | 294 | 9 | 204 | Yes |
| 5 | Gene | 17892 | 36 | 68 | 9868 | Yes |
| 6 | Movie | 7755 | 1869 | 5 | 6045 | Yes |
| 7 | M2B | 1240 | 250 | 5 | 799 | Yes |
| 8 | SCUT | 1500 | 300 | 5 | 879 | Yes |
| 9 | fbp5500 | 5500 | 512 | 5 | 362 | Yes |
| 10 | RAF-ML | 4908 | 200 | 6 | 3120 | Yes |
| 11 | Twitter | 10040 | 200 | 8 | 7802 | Yes |
| 12 | Flickr | 11150 | 200 | 8 | 4978 | Yes |

constant value, providing a base for stable learning.

$$||\hat{\beta} - \bar{\mathbf{W}}||^2 = O(1) + O(n^{-1/2})\sqrt{\mathbf{E}_{\mathbf{X}\sim\mathbb{N}}w(\mathbf{X})^2\sigma}, \quad (7)$$

where $\mathbb{N}$ denotes the Gaussian distribution.

In this paper, we use an automatic differentiation framework (PyTorch) to run Eq. 5 on an RTX3090 GPU shader with 24G RAM. Note that since the feature spaces of Gene, Twitter, and Flicker are vast, we split the batch to conduct the learning of weights $\hat{\beta}$. Here, since the split-batch implementation of these datasets, the method cannot be directly globally modeled, and for this reason, we train on these three datasets to conduct more epochs (training rounds).

So far, we observe a phenomenon that the sample space reconstructed by the weight assignor is overly compact for the downstream learners, and these learners underperform on the test samples. To solve this problem, we introduce some positive incentive noise. The source of these positive incentive noises is the doubtful samples ($\mathcal{B} - \hat{\mathcal{B}}$) after being filtered by the prototype guidance.

The samples weighted by our algorithm occupy only 50-90% of the raw samples, as shown in Table 1. Although the label space of the remaining samples has a high probability of carrying noise, noisy data are not always noxious and may have positive incentive properties [Li, 2022]. We want to assign a certain amount of weight to these samples to model the robust decision boundary without disrupting the generalization ability of the model as much as possible. Here, we use a customized normal distribution $0.01 \times \mathbb{N}(0, 1)$ which randomly assigns weights to these samples. Through the experimental part, we observe that this strategy makes the prediction of the label distribution more relaxed, and this method plays the role of regularization due to the feature decoupling that makes the prediction result of the LDL model overly compact.

## 5 EXPERIMENTS

**Algorithm configurations.** We conduct experiments on 12 datasets and the characteristics of the datasets are summarized in Table 1. Except for dataset wc-LDL, other data sets are from [Geng, 2016, Lyons et al., 1998]. This new release

Table 2: The performance of our proposed method with the comparison algorithms on 12 datasets. The best-performing results are marked in **bold**.

| Dataset | Algorithm | Chebyshev ↓ | Clark ↓ | Canberra ↓ | K-L ↓ | Cosine ↑ | Intersection ↑ |
|---|---|---|---|---|---|---|---|
| wc-LDL | Ours | **0.0743 ± 0.0011** | **0.3884 ± 0.0055** | **0.7667 ± 0.0033** | 0.0421 ± 0.0008 | **0.9896 ± 0.0009** | **0.8813 ± 0.0014** |
| | Baseline-LDL | 0.0788 ± 0.0019 | 0.4008 ± 0.0042 | 0.7770 ± 0.0023 | 0.0408 ± 0.0056 | 0.9801 ± 0.0017 | 0.8760 ± 0.0015 |
| | INP | 0.0779 ± 0.0021 | 0.3980 ± 0.0051 | 0.7779 ± 0.0030 | **0.0404 ± 0.0020** | 0.9883 ± 0.0009 | 0.8778 ± 0.0014 |
| | PCA | 0.0748 ± 0.0122 | 0.4008 ± 0.0020 | 0.7883 ± 0.0012 | 0.0422 ± 0.0051 | 0.9887 ± 0.0012 | 0.8790 ± 0.0034 |
| | LDL-LRR | 0.0923 ± 0.0030 | 0.4212 ± 0.0036 | 0.8135 ± 0.0024 | 0.0511 ± 0.0049 | 0.9718 ± 0.0022 | 0.8669 ± 0.0047 |
| | LDL-LCLR | 0.1057 ± 0.0019 | 1.0569 ± 0.0039 | 0.7890 ± 0.0039 | 0.0545 ± 0.0037 | 0.9668 ± 0.0049 | 0.8383 ± 0.0018 |
| | LDLSF | 0.1009 ± 0.0038 | 0.4199 ± 0.0044 | 0.9008 ± 0.0015 | 0.0519 ± 0.0040 | 0.9779 ± 0.0018 | 0.8660 ± 0.0022 |
| | LALOT | 0.0989 ± 0.0019 | 0.6689 ± 0.0019 | 0.8089 ± 0.0049 | 0.0477 ± 0.0018 | 0.9476 ± 0.0020 | 0.8700 ± 0.0033 |
| | BFGS-LLD | 0.1122 ± 0.0039 | 1.5657 ± 0.0021 | 0.7998 ± 0.0020 | 0.0498 ± 0.0051 | 0.9704 ± 0.0036 | 0.8611 ± 0.0016 |
| JAFFE | Ours | **0.0822 ± 0.0019** | **0.4001 ± 0.0033** | **0.7888 ± 0.0043** | 0.4053 ± 0.0013 | **0.9891 ± 0.0002** | **0.8846 ± 0.0055** |
| | Baseline-LDL | 0.0899 ± 0.0033 | 0.4128 ± 0.0027 | 0.8007 ± 0.0013 | 0.4212 ± 0.0074 | 0.9709 ± 0.0013 | 0.8699 ± 0.0015 |
| | INP | 0.0854 ± 0.0018 | 0.4008 ± 0.0030 | 0.7955 ± 0.0023 | **0.4010 ± 0.0012** | 0.9799 ± 0.0014 | 0.8809 ± 0.0015 |
| | PCA | 0.0832 ± 0.0033 | 0.4012 ± 0.0008 | 0.7910 ± 0.0043 | 0.4155 ± 0.0087 | 0.9823 ± 0.0049 | 0.8832 ± 0.0055 |
| | LDL-LRR | 0.0866 ± 0.0021 | 0.4220 ± 0.0036 | 0.8001 ± 0.0024 | 0.4258 ± 0.0049 | 0.9610 ± 0.0022 | 0.8689 ± 0.0047 |
| | LDL-LCLR | 0.1057 ± 0.0019 | 1.0569 ± 0.0039 | 0.7890 ± 0.0039 | 0.5045 ± 0.0037 | 0.9668 ± 0.0049 | 0.8383 ± 0.0018 |
| | LDLSF | 0.1122 ± 0.0038 | 0.4397 ± 0.0044 | 0.9212 ± 0.0015 | 0.5557 ± 0.0040 | 0.9779 ± 0.0018 | 0.8660 ± 0.0022 |
| | LALOT | 0.0979 ± 0.0018 | 0.6799 ± 0.0021 | 0.8077 ± 0.0039 | 0.4756 ± 0.0015 | 0.9433 ± 0.0111 | 0.8423 ± 0.0034 |
| | BFGS-LLD | 0.1334 ± 0.0139 | 1.6648 ± 0.0023 | 0.7999 ± 0.0022 | 0.4771 ± 0.0051 | 0.9711 ± 0.0036 | 0.8655 ± 0.0116 |
| SBU | Ours | **0.0811 ± 0.0023** | **0.3987 ± 0.0024** | **0.7533 ± 0.0027** | **0.0354 ± 0.0031** | **0.9888 ± 0.0066** | **0.8997 ± 0.0030** |
| | Baseline-LDL | 0.0970 ± 0.0442 | 0.4151 ± 0.0088 | 0.7810 ± 0.0023 | 0.0414 ± 0.0019 | 0.9711 ± 0.0013 | 0.8797 ± 0.0016 |
| | INP | 0.0833 ± 0.0020 | 0.3994 ± 0.0010 | 0.7611 ± 0.0020 | 0.0365 ± 0.0014 | 0.9811 ± 0.0015 | 0.8900 ± 0.0017 |
| | PCA | 0.0820 ± 0.0045 | 0.3999 ± 0.0011 | 0.7689 ± 0.0111 | 0.0370 ± 0.0077 | 0.9866 ± 0.0015 | 0.8953 ± 0.0044 |
| | LDL-LRR | 0.0912 ± 0.0036 | 0.4013 ± 0.0039 | 0.7602 ± 0.0021 | 0.0369 ± 0.0028 | 0.9697 ± 0.0029 | 0.8891 ± 0.0033 |
| | LDL-LCLR | 0.1100 ± 0.0025 | 0.9660 ± 0.0039 | 0.7897 ± 0.0033 | 0.0511 ± 0.0021 | 0.9677 ± 0.0056 | 0.8555 ± 0.0032 |
| | LDLSF | 0.1009 ± 0.0038 | 0.4199 ± 0.0044 | 0.9008 ± 0.0015 | 0.0519 ± 0.0040 | 0.9780 ± 0.0029 | 0.8660 ± 0.0022 |
| | LALOT | 0.0899 ± 0.0021 | 0.6563 ± 0.0019 | 0.8132 ± 0.0100 | 0.0468 ± 0.0021 | 0.9441 ± 0.0011 | 0.8723 ± 0.0034 |
| | BFGS-LLD | 0.1119 ± 0.0030 | 1.4657 ± 0.0022 | 0.7700 ± 0.0025 | 0.0492 ± 0.0053 | 0.9753 ± 0.0036 | 0.8710 ± 0.0019 |
| Scene | Ours | **0.2981 ± 0.0024** | **2.3077 ± 0.0013** | **6.4133 ± 0.0029** | **0.8029 ± 0.0020** | **0.7991 ± 0.0011** | **0.5699 ± 0.0014** |
| | Baseline-LDL | 0.3155 ± 0.0022 | 2.3559 ± 0.0155 | 6.6958 ± 0.1231 | 0.8533 ± 0.0099 | 0.7664 ± 0.0015 | 0.5349 ± 0.0014 |
| | INP | 0.2998 ± 0.0020 | 2.3374 ± 0.0018 | 6.5163 ± 0.0018 | 0.8111 ± 0.0029 | 0.7890 ± 0.0049 | 0.5691 ± 0.0010 |
| | PCA | 0.3010 ± 0.0213 | 2.3266 ± 0.0085 | 6.533 ± 0.0091 | 0.8097 ± 0.0031 | 0.7913 ± 0.0033 | 0.5612 ± 0.0006 |
| | LDL-LRR | 0.2989 ± 0.0111 | 2.3698 ± 0.0051 | 6.4777 ± 0.0025 | 0.8362 ± 0.0069 | 0.7744 ± 0.0077 | 0.5444 ± 0.0049 |
| | LDL-LCLR | 0.3740 ± 0.0066 | 2.4986 ± 0.0066 | 6.8600 ± 0.0067 | 0.8559 ± 0.0039 | 0.7119 ± 0.0122 | 0.5119 ± 0.0081 |
| | LDLSF | 0.3441 ± 0.0249 | 2.9884 ± 0.0055 | 6.6900 ± 0.0055 | 0.8391 ± 0.0044 | 0.7336 ± 0.0088 | 0.5660 ± 0.0041 |
| | LALOT | 0.3129 ± 0.0152 | 2.3999 ± 0.0044 | 6.6366 ± 0.0078 | 0.8226 ± 0.0033 | 0.7390 ± 0.0100 | 0.5224 ± 0.0066 |
| | BFGS-LLD | 0.3598 ± 0.0020 | 2.4998 ± 0.0033 | 6.7999 ± 0.0049 | 0.8400 ± 0.0033 | 0.7333 ± 0.0064 | 0.5199 ± 0.0055 |
| Gene | Ours | **0.0480 ± 0.0033** | **2.1008 ± 0.0259** | **14.0800 ± 0.0153** | **0.2320 ± 0.0094** | **0.8406 ± 0.0023** | **0.7997 ± 0.0077** |
| | Baseline-LDL | 0.0509 ± 0.0066 | 2.2004 ± 0.0055 | 14.1449 ± 0.2448 | 0.2440 ± 0.0024 | 0.8345 ± 0.0009 | 0.7821 ± 0.0016 |
| | INP | 0.0488 ± 0.0012 | 2.1029 ± 0.0259 | 14.0888 ± 0.0551 | 0.2335 ± 0.0044 | 0.8395 ± 0.0032 | 0.7984 ± 0.0066 |
| | PCA | 0.0482 ± 0.0013 | 2.1020 ± 0.0212 | 14.0835 ± 0.0142 | 0.2321 ± 0.0087 | 0.8390 ± 0.0016 | 0.7989 ± 0.0099 |
| | LDL-LRR | 0.0494 ± 0.0039 | 2.1888 ± 0.0861 | 14.2550 ± 0.0144 | 0.2400 ± 0.0077 | 0.8388 ± 0.0144 | 0.7789 ± 0.0040 |
| | LDL-LCLR | 0.0511 ± 0.0022 | 2.2201 ± 0.0444 | 14.2101 ± 0.0510 | 0.2566 ± 0.0047 | 0.8302 ± 0.0012 | 0.7722 ± 0.0060 |
| | LDLSF | 0.0513 ± 0.0030 | 2.2221 ± 0.0036 | 14.3667 ± 0.0265 | 0.2445 ± 0.0077 | 0.8320 ± 0.0010 | 0.7701 ± 0.0026 |
| | LALOT | 0.0505 ± 0.0033 | 2.1989 ± 0.0194 | 14.1855 ± 0.0922 | 0.2443 ± 0.0088 | 0.8297 ± 0.0060 | 0.7888 ± 0.0013 |
| | BFGS-LLD | 0.0578 ± 0.0066 | 2.3008 ± 0.0188 | 14.3559 ± 0.1556 | 0.2480 ± 0.0015 | 0.8300 ± 0.0049 | 0.7786 ± 0.0070 |
| Movie | Ours | **0.1071 ± 0.0008** | **0.4997 ± 0.0064** | **0.9710 ± 0.0044** | **0.0970 ± 0.0008** | **0.9595 ± 0.0063** | 0.8791 ± 0.0019 |
| | Baseline-LDL | 0.1109 ± 0.0033 | 0.5119 ± 0.0155 | 1.0889 ± 0.0111 | 0.1355 ± 0.0022 | 0.9422 ± 0.0333 | 0.8744 ± 0.0054 |
| | INP | 0.1089 ± 0.0018 | 0.5001 ± 0.0044 | 0.9722 ± 0.0040 | 0.0977 ± 0.0008 | 0.9585 ± 0.0061 | **0.8861 ± 0.0006** |
| | PCA | 0.1077 ± 0.0006 | 0.5013 ± 0.0032 | 0.9720 ± 0.0032 | 0.0972 ± 0.0005 | 0.9590 ± 0.0002 | 0.8853 ± 0.0022 |
| | LDL-LRR | 0.1107 ± 0.0009 | 0.5019 ± 0.0010 | 0.9801 ± 0.0061 | 0.1045 ± 0.0049 | 0.9591 ± 0.0022 | 0.8772 ± 0.0027 |
| | LDL-LCLR | 0.1177 ± 0.0086 | 0.5345 ± 0.0040 | 1.1533 ± 0.0111 | 0.1559 ± 0.0030 | 0.9360 ± 0.0049 | 0.8222 ± 0.0011 |
| | LDLSF | 0.1155 ± 0.0045 | 0.5339 ± 0.0062 | 1.1152 ± 0.0050 | 0.1540 ± 0.0041 | 0.9445 ± 0.0020 | 0.8551 ± 0.0044 |
| | LALOT | 0.1221 ± 0.0110 | 0.5440 ± 0.0033 | 1.1112 ± 0.0040 | 0.1503 ± 0.0008 | 0.9477 ± 0.0022 | 0.8559 ± 0.0002 |
| | BFGS-LLD | 0.1310 ± 0.0032 | 0.5230 ± 0.0022 | 1.1170 ± 0.0024 | 0.1595 ± 0.0155 | 0.9400 ± 0.0003 | 0.8491 ± 0.0018 |

dataset (wc-LDL) has 500 watercolor images and corresponding label distribution (12 emotions). For the wc-LDL dataset, we give 12 emotion tips to the annotators including 5 men and 5 women. Finally, the outputs of these 10 annotators are normalized as the label distribution corresponding to the image. Note that before watercolor images are annotated, we ask these experts to take a comprehensive view based on the lines and color combinations of the images. For example, dense lines express vexation, blue denotes depression, and red denotes enthusiasm. To construct a training set with noisy labels, we use a switching algorithm with randomness at 20-35% of the training set. This algorithm aims to exchange the values of the label distribution over a label distribution ([0.1, 0.2, 0.7] → [0.7, 0.1, 0.2]). We develop a simple linear model with a data pre-processing method (Ours). In addition, we set up a baseline (Baseline-LDL) with data pre-processing (without the strategy of randomly assigning weights to noisy samples). To evaluate the performance of LDL models, we use the six metrics proposed by [Geng, 2016], including Chebyshev distance ↓, Clark

distance ↓, Canberra distance ↓, KL divergence ↓, Cosine similarity ↑, and Intersection similarity ↑. ↓ represents the indicator's performance favoring low values and ↑ represents the indicator's performance favoring high values.

**Experimental setting.** We conduct comparative experiments with seven LDL algorithms (Baseline-LDL, INP [Zheng and Jia, 2022], BFGS-LLD [Geng, 2016], LDL-LRR [Jia et al., 2021], LDL-LCLR [Ren et al., 2019b], LDLSF [Ren et al., 2019a], principal component analysis (PCA) and LALOT [Zhao and Zhou, 2018]). Baseline-LDL as one of the methods of comparison is trained only on a relatively clean set of samples. INP presents an implicit representation to estimate the uncertainty of the label space. BFGS-LLD is based on a linear model, the loss function is K-L divergence, and the optimization method is the quasi-Newton approach. LDL-LRR and LDL-LCLR both consider label correlations in the learning process, with the former considering the order relationship of the labels and the latter capturing global relationships between labels. For LDL-LRR, the parameters $\lambda$ and $\beta$ are tuned from $10^{\{-6,-5,...,-2,-1\}}$ and $10^{\{-3,-2,...,1,2\}}$, respectively. For LDL-LCLR, the parameters $\lambda_1, \lambda_2, \lambda_3, \lambda_4$ and $k$ are set to $0.0001, 0.001, 0.001, 0.001$ and $4$, respectively. LDLSF leverages label-specific features and common features simultaneously, whose parameters $\lambda_1, \lambda_2$ and $\lambda_3$ are tuned from $10^{\{-6,-5,...,-2,-1\}}$, respectively, and $\rho$ is set to $10^{-3}$. LALOT adopts optimal transport distance as the loss function, and the trade-off parameter $C$ and the regularization coefficient $\lambda$ are set to 200 and 0.2, respectively. The fine-tuning settings for all comparison methods are referenced in [Jia et al., 2021]. In addition to the above comparison algorithms, we introduce PCA as one of the comparison algorithms because PCA also serves to decouple the feature space. PCA serves as a preprocessing framework (retaining 80 percent of the features), followed immediately by a standard linear regressor.

**Results and analysis.** We conduct 10 times 5-fold cross-validation on each dataset. The experimental results are presented in the form of "mean±std" in Table 2 (the rest of the showcase is released in the supplemental material). Overall, our proposed method outperforms other comparison algorithms on all evaluation metrics. Three main reasons contribute to the competitive results of our approach. **i):** With the uniform optimization scheme, our algorithm performs better than the baseline algorithm (Baseline-LDL) due to the feature decoupling. **ii):** From the performance of the baseline model, our method obtains competitive results on most of the metrics, thanks to the samples with uncertainty. In addition, we note that methods with label constraints (e.g., LDL-LRR) also perform well, and it may be due to label constraints that ignore the noisy label interference. **iii):** Since the powerful learning capability of kernel mapping, the advantage of our approach is vast on

image and text datasets. Moreover, we evaluate the range of p-values for the six metrics on 12 data sets.

Chebyshev $[1.54e-104, 1.00e+00]$, Clark $[5.44e-97, 1.98e-02]$, Canberra $[9.62e-98, 1.10e-01]$, K-L $[1.77e-102, 1.99e-01]$, C osine $[1.33e-99, 2.01e-01]$, and Intersection $[1.33e-113, 7.88e-01]$

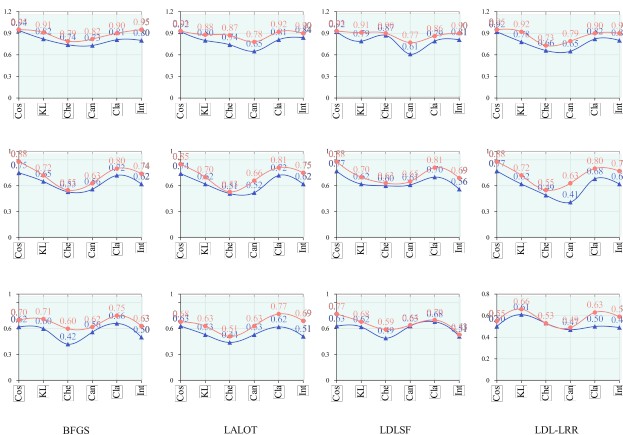

Figure 4: This figure shows the performance comparison of these four algorithms after implementing our framework. The first row indicates that the algorithms are run on the Movie dataset with label noise, the second row indicates that the algorithms are run on the M2B dataset with label noise, and the third row indicates that the algorithms are run on the SCUT dataset with label noise.

**Parameter sensitivity analysis.** Our method has two parameters, including the regularization parameter $\lambda$ and $\gamma$. To analyze the sensitivity of $\lambda$ and $\gamma$, we run our method with two sets $\{0.001, 0.005, 0.01, 0.05, 0.1\}$, and $\{0.001, 0.005, 0.01, 0.05, 0.1\}$ on the Gene dataset. We conduct a 5-fold cross-validation and achieved the following results (Cosine ↑): $\lambda_i \rightarrow \{0.9841 \pm 0.0133, 0.9802 \pm 0.0007, \textbf{0.9896} \pm \textbf{0.0009}, 0.9876 \pm 0.0044, 0.9877 \pm 0.0084\}$. $\gamma_i \rightarrow \{0.9813 \pm 0.0043, 0.9821 \pm 0.0075, \textbf{0.9896} \pm \textbf{0.0009}, 0.9890 \pm 0.0100, 0.9892 \pm 0.0023\}$.

**Ablation study.** To demonstrate the effectiveness of the loss function and the module of our model, we conduct an ablation study involving the following three experiments, and the results are shown in Table 3: **(a)** w/o feature decoupling: We remove the weight assignment strategy, and our model is trained only on samples that are weighted by clean samples and those with uncertainty. **(b)** w/o the kernel mapping: We remove the loss term in Eq. 5 for the kernel trick, keeping only the first loss term and a regularization term. **(c)** w/o regularization term: We remove the third loss term in Eq. 5. We conduct 10 times 5-fold cross-validation on the dataset of the ablation experiment.

**Discussion.** The stability-trust framework can offer a more compact regression space with the help of prototype learn-

Table 3: Ablation study. Effectiveness of the loss functions and the modules on Gene. Quantitative results demonstrate the effectiveness of each module.

| Algorithm | Chebyshev ↓ | Clark ↓ | Canberra ↓ | K-L ↓ | Cosine ↑ | Intersection ↑ |
|---|---|---|---|---|---|---|
| Ours | 0.0480 ± 0.0033 | 2.1008 ± 0.0259 | 14.0800 ± 0.0153 | 0.2320 ± 0.0094 | 0.8406 ± 0.0023 | 0.7997 ± 0.0077 |
| w/o FD | 0.0499 ± 0.0063 | 2.1331 ± 0.0220 | 14.1866 ± 0.0155 | 0.2329 ± 0.0110 | 0.8361 ± 0.0023 | 0.7900 ± 0.0056 |
| w/o KT | 0.0511 ± 0.0034 | 2.1226 ± 0.0230 | 14.1164 ± 0.0163 | 0.2445 ± 0.0011 | 0.8389 ± 0.0023 | 0.7884 ± 0.0039 |
| w/o RT | 0.0498 ± 0.0019 | 2.1121 ± 0.0100 | 14.2911 ± 0.0156 | 0.2333 ± 0.0094 | 0.8398 ± 0.0022 | 0.7990 ± 0.0075 |

Table 4: Overall performance of *MedMNIST* (v2) in metrics of AUC and ACC, using ResNet-18 / ResNet-50 [Al-Haija and Adebanjo, 2020] with resolution 28 and 224, auto-sklearn, AutoKeras, Google AutoML Vision, FPVT [Liu et al., 2022], and Ours.

| Methods | PathMNIST | | ChestMNIST | | DermaMNIST | | OCTMNIST | | PneumoniaMNIST | | BloodMNIST | |
|---|---|---|---|---|---|---|---|---|---|---|---|---|
| | AUC | ACC | AUC | ACC | AUC | ACC | AUC | ACC | AUC | ACC | AUC | ACC |
| ResNet-18 (28) | 0.970 | 0.823 | 0.700 | 0.941 | 0.846 | 0.711 | 0.950 | 0.730 | 0.953 | 0.840 | 0.990 | 0.932 |
| ResNet-18 (224) | 0.971 | 0.860 | 0.702 | 0.943 | 0.890 | 0.721 | 0.952 | 0.753 | 0.960 | 0.848 | 0.990 | 0.955 |
| ResNet-50 (28) | 0.971 | 0.833 | 0.691 | 0.942 | 0.883 | 0.705 | 0.923 | 0.744 | 0.941 | 0.833 | 0.989 | 0.950 |
| ResNet-50 (224) | 0.973 | 0.841 | 0.676 | 0.929 | 0.890 | 0.713 | 0.944 | 0.702 | 0.960 | 0.893 | 0.972 | 0.935 |
| auto-sklearn | 0.444 | 0.386 | 0.640 | 0.625 | 0.886 | 0.730 | 0.843 | 0.591 | 0.940 | 0.863 | 0.982 | 0.870 |
| AutoKeras | 0.951 | 0.860 | 0.711 | 0.932 | 0.910 | 0.755 | 0.950 | 0.731 | 0.965 | 0.911 | 0.994 | 0.950 |
| Google AutoML Vision | 0.981 | 0.833 | 0.710 | 0.941 | 0.920 | 0.749 | 0.932 | 0.722 | 0.990 | 0.930 | 0.992 | 0.957 |
| FPVT | 0.965 | 0.900 | 0.715 | 0.940 | 0.911 | 0.753 | 0.952 | 0.769 | 0.930 | 0.892 | 0.970 | 0.942 |
| Ours | **0.992** | **0.939** | **0.816** | **0.959** | **0.931** | **0.826** | **0.966** | **0.835** | **0.988** | **0.960** | **0.992** | **0.989** |

ing. To verify our theory, we use t-SNE to enforce the predicted label distributions and the ground-truth label distributions of the raw dataset, respectively. We evaluate four datasets (Movie, M2B, SCUT, and fbp5500) and the results are visualized in the supplemental material. We note that the stability-trust framework can aggregate similar label vectors more compactly. Although a compact prediction space can reduce the number of outliers, this results in a loss of accuracy in quantitative evaluation. In this paper, we propose to leverage the rest of the samples with noise to give them small weights as the training set to alleviate this problem. To evaluate the effectiveness of this method, we propose a metric that computes the average distance between the predicted label distribution to the prototype vector, which is written as:

$$\text{Score} = \text{Sigmoid}(\frac{1}{N}\sum_{i}^{N}||\mathcal{L}_i - p||_2), \qquad (8)$$

where Sigmoid is intended to normalize the output. We use our algorithm to evaluate the above problem on four data sets (Movie, M2B, SCUT, and fbp5500). The score of our algorithm is {0.35, 0.42, 0.44, 0.29} when trained on only clean samples, and scores trained on samples containing noise are {0.47, 0.45, 0.51, 0.33}; respectively. The training set containing noisy samples can be a more relaxed prediction result, which theoretically extends the decision boundary of the model.

In addition, we need to evaluate whether the stability-trust framework is suited for the multi-classification tasks (MedMNIST (v2) Yang et al. [2023]). Here, this framework is evaluated only on the multi-class dataset.

We use ResNet-18 as the baseline method. At first, we used

a label enhancement algorithm [Xu et al., 2019] to convert *MedMNIST* (v2) into a pseudo-LDL dataset. We use ResNet-18 [Ayyachamy et al., 2019] set as the baseline method. We use cross-entropy and set the batch size to 128 during the model training. We utilize an AdamW optimizer with an initial learning rate of 0.001 and train the network for 100 epochs, delaying the learning rate by 0.1 after 50 and 75 epochs. This ResNet-18 is implemented on *MedMNIST* (v2) after being enforced by the stability-trust framework. It is worth noting that the image is flattened and then input to the stability-trust framework. As shown in Table 4, our method achieves optimal results on noisy labels' datasets (10% noise). Besides, we demonstrate the degree of performance improvement through BFGS-LLD, LALOT, LDLSF, and LDL-LRR algorithms conducted on Movie, M2B, and SCUT datasets. These methods used our framework to pre-process the dataset before implementation. As shown in Figure 4, the stability-trust framework as a data pre-processing technique can enable the performance of the LDL algorithm to be enhanced on a benchmark with noise.

## 6 CONCLUSION

We propose a stability-trust framework to overcome the problem of noisy labels on 13 benchmarks (12 label distributions and 1 multi-classification). Our approach has two key components, one is prototype learning to guide the model to learn the compact space; the other is the feature decoupling strategy. Our method is more efficient compared to the existent LDL de-noising methods and it does not require additional knowledge and an expensive sampling process. A large number of experimental results demonstrate the effectiveness of our approach.

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

# Trusted re-weighting for label distribution learning
# (Supplementary Material)

**Zhuoran Zheng**[1]     **Chen Wu**[2]     **Yeying Jin**[3]     **Xiuyi Jia**[*1]

[1]School of Computer Science and Engineering, Nanjing University of Science and Technology, Nanjing, China
[2]University of Science and Technology of China, Hefei, China
[3]Department of Electrical and Computer Engineering, National University of Singapore, Singapore

## OVERVIEW

In this supplemental material, we introduce the implementation details of the experiment in Section I. We show a heat map for our algorithm to eliminate correlations between features in Section II.

## 7   IMPLEMENTATION OF EXPERIMENT

We conduct 10 times 5-fold cross-validation on each dataset. The experimental results are presented in the form of "mean±std" in Tables 5 and 6. Our algorithm achieves competitive results compared to other algorithms.

The stability-trust framework (no positive incentive noise) can offer a more compact regression space with the help of prototype learning. However, our algorithm with positive incentive noise can extend the decision boundary. To verify our theory, we use t-SNE [Van Der Maaten, 2014] to enforce the predicted label distributions and the ground-truth label distributions of the raw dataset, respectively. We evaluate four datasets (Movie, M2B, SCUT, and fbp5500) and the results are visualized in Figure 5.

## 8   HEAT MAPS OF FEATURE CORRELATIONS

We use heat maps to evaluate feature correlations in the Gene dataset to verify that our method has the capability of attribute decoupling (see Figure 6). Figure 6(a) demonstrates strong correlation between the raw dataset features and Figure 6(b) demonstrates weak correlation between the features.

---

[*]Correspondence. This work was supported by the National Natural Science Foundation of China (62176123).
[*]Correspondence. This work was supported by the National Natural Science Foundation of China (62176123).

*Accepted for the 40[th] Conference on Uncertainty in Artificial Intelligence* (UAI 2024).

Table 5: The performance of our proposed method with the comparison algorithms on 12 datasets. The best-performing results are marked in **bold**.

| Dataset | Algorithm | Chebyshev ↓ | Clark ↓ | Canberra ↓ | K-L ↓ | Cosine ↑ | Intersection ↑ |
|---|---|---|---|---|---|---|---|
| Gene | Ours | **0.0480 ± 0.0033** | **2.1008 ± 0.0259** | **14.0800 ± 0.0153** | **0.2320 ± 0.0094** | **0.8406 ± 0.0023** | **0.7997 ± 0.0077** |
| | Baseline-LDL | 0.0509 ± 0.0066 | 2.2004 ± 0.0055 | 14.1449 ± 0.2448 | 0.2440 ± 0.0024 | 0.8345 ± 0.0009 | 0.7821 ± 0.0016 |
| | INP | 0.0488 ± 0.0012 | 2.1029 ± 0.0259 | 14.0888 ± 0.0551 | 0.2335 ± 0.0044 | 0.8395 ± 0.0032 | 0.7984 ± 0.0066 |
| | PCA | 0.0482 ± 0.0013 | 2.1020 ± 0.0212 | 14.0835 ± 0.0142 | 0.2321 ± 0.0087 | 0.8390 ± 0.0016 | 0.7989 ± 0.0099 |
| | LDL-LRR | 0.0494 ± 0.0039 | 2.1888 ± 0.0861 | 14.2550 ± 0.0144 | 0.2400 ± 0.0077 | 0.8388 ± 0.0144 | 0.7789 ± 0.0040 |
| | LDL-LCLR | 0.0511 ± 0.0022 | 2.2201 ± 0.0444 | 14.2101 ± 0.0510 | 0.2566 ± 0.0047 | 0.8302 ± 0.0012 | 0.7722 ± 0.0060 |
| | LDLSF | 0.0513 ± 0.0030 | 2.2221 ± 0.0036 | 14.3667 ± 0.0265 | 0.2445 ± 0.0077 | 0.8320 ± 0.0010 | 0.7701 ± 0.0026 |
| | LALOT | 0.0505 ± 0.0033 | 2.1989 ± 0.0194 | 14.1855 ± 0.0922 | 0.2443 ± 0.0088 | 0.8297 ± 0.0060 | 0.7888 ± 0.0013 |
| | BFGS-LLD | 0.0578 ± 0.0066 | 2.3008 ± 0.0188 | 14.3559 ± 0.1556 | 0.2480 ± 0.0015 | 0.8300 ± 0.0049 | 0.7786 ± 0.0070 |
| Movie | Ours | **0.1071 ± 0.0008** | **0.4997 ± 0.0064** | **0.9710 ± 0.0044** | **0.0970 ± 0.0008** | **0.9595 ± 0.0063** | 0.8791 ± 0.0019 |
| | Baseline-LDL | 0.1109 ± 0.0033 | 0.5119 ± 0.0155 | 1.0889 ± 0.0111 | 0.1355 ± 0.0022 | 0.9422 ± 0.0333 | 0.8744 ± 0.0054 |
| | INP | 0.1089 ± 0.0018 | 0.5001 ± 0.0044 | 0.9722 ± 0.0040 | 0.0977 ± 0.0008 | 0.9585 ± 0.0061 | **0.8861 ± 0.0006** |
| | PCA | 0.1077 ± 0.0006 | 0.5013 ± 0.0032 | 0.9720 ± 0.0032 | 0.0972 ± 0.0005 | 0.9590 ± 0.0002 | 0.8853 ± 0.0022 |
| | LDL-LRR | 0.1107 ± 0.0009 | 0.5019 ± 0.0010 | 0.9801 ± 0.0061 | 0.1045 ± 0.0049 | 0.9591 ± 0.0022 | 0.8772 ± 0.0027 |
| | LDL-LCLR | 0.1177 ± 0.0086 | 0.5345 ± 0.0040 | 1.1533 ± 0.0111 | 0.1559 ± 0.0030 | 0.9360 ± 0.0049 | 0.8222 ± 0.0011 |
| | LDLSF | 0.1155 ± 0.0045 | 0.5339 ± 0.0062 | 1.1152 ± 0.0050 | 0.1540 ± 0.0041 | 0.9445 ± 0.0020 | 0.8551 ± 0.0044 |
| | LALOT | 0.1221 ± 0.0110 | 0.5440 ± 0.0033 | 1.1112 ± 0.0040 | 0.1503 ± 0.0008 | 0.9477 ± 0.0022 | 0.8559 ± 0.0002 |
| | BFGS-LLD | 0.1310 ± 0.0032 | 0.5230 ± 0.0022 | 1.1170 ± 0.0024 | 0.1595 ± 0.0155 | 0.9400 ± 0.0003 | 0.8491 ± 0.0018 |
| M2B | Ours | **0.3691 ± 0.0021** | **1.1541 ± 0.0131** | **2.0880 ± 0.0056** | **0.4872 ± 0.0026** | **0.8028 ± 0.0033** | **0.6800 ± 0.0082** |
| | Baseline-LDL | 0.3997 ± 0.0077 | 1.2889 ± 0.0056 | 2.1992 ± 0.2887 | 0.5006 ± 0.0044 | 0.7887 ± 0.0099 | 0.6558 ± 0.0065 |
| | INP | 0.3763 ± 0.0022 | 1.1560 ± 0.0102 | 2.0889 ± 0.0055 | 0.4880 ± 0.0023 | 0.7998 ± 0.0022 | 0.6703 ± 0.0033 |
| | PCA | 0.3731 ± 0.0017 | 1.1555 ± 0.0123 | 2.0893 ± 0.0048 | 0.4883 ± 0.0112 | 0.7999 ± 0.0091 | 0.6745 ± 0.0044 |
| | LDL-LRR | 0.3793 ± 0.0010 | 1.1590 ± 0.0167 | 2.1084 ± 0.0034 | 0.4998 ± 0.0012 | 0.7931 ± 0.0023 | 0.6634 ± 0.0077 |
| | LDL-LCLR | 0.4040 ± 0.0082 | 1.2444 ± 0.0045 | 2.2000 ± 0.0009 | 0.4996 ± 0.0013 | 0.7760 ± 0.0079 | 0.6555 ± 0.0012 |
| | LDLSF | 0.4159 ± 0.0055 | 1.3105 ± 0.0041 | 2.2155 ± 0.0076 | 0.5002 ± 0.0006 | 0.7552 ± 0.0004 | 0.6234 ± 0.0033 |
| | LALOT | 0.3881 ± 0.0099 | 1.4883 ± 0.0012 | 2.1257 ± 0.0268 | 0.4990 ± 0.0008 | 0.7549 ± 0.0021 | 0.6620 ± 0.0053 |
| | BFGS-LLD | 0.3811 ± 0.0044 | 1.3650 ± 0.0002 | 2.1992 ± 0.0095 | 0.4995 ± 0.0005 | 0.7699 ± 0.0040 | 0.6532 ± 0.0009 |
| SCUT | Ours | **0.3851 ± 0.0034** | 1.2580 ± 0.0191 | **2.1901 ± 0.0042** | **0.4900 ± 0.0036** | **0.7007 ± 0.0002** | **0.6955 ± 0.0004** |
| | Baseline-LDL | 0.4008 ± 0.0008 | 1.3365 ± 0.0155 | 2.2110 ± 0.0339 | 0.5119 ± 0.0044 | 0.6697 ± 0.0012 | 0.6489 ± 0.0055 |
| | INP | 0.3895 ± 0.0021 | 1.2640 ± 0.0111 | 2.1995 ± 0.0095 | 0.4911 ± 0.0030 | 0.6990 ± 0.0002 | 0.6904 ± 0.0001 |
| | PCA | 0.3903 ± 0.0023 | 1.2642 ± 0.0155 | 2.1942 ± 0.0044 | 0.4903 ± 0.0017 | 0.6992 ± 0.0001 | 0.6912 ± 0.0022 |
| | LDL-LRR | 0.3901 ± 0.0011 | 1.3000 ± 0.0122 | 2.2006 ± 0.0039 | 0.5088 ± 0.0026 | 0.6992 ± 0.0023 | 0.6889 ± 0.0007 |
| | LDL-LCLR | 0.4240 ± 0.0042 | 1.3444 ± 0.0055 | 2.2450 ± 0.0016 | 0.5131 ± 0.0022 | 0.6261 ± 0.0005 | 0.5500 ± 0.0012 |
| | LDLSF | 0.4360 ± 0.0015 | **1.2185 ± 0.0022** | 2.2159 ± 0.0076 | 0.5120 ± 0.0006 | 0.6261 ± 0.0004 | 0.5534 ± 0.0030 |
| | LALOT | 0.3999 ± 0.0009 | 1.4983 ± 0.0012 | 2.2207 ± 0.0158 | 0.4995 ± 0.0002 | 0.6549 ± 0.0020 | 0.6411 ± 0.0044 |
| | BFGS-LLD | 0.3992 ± 0.0055 | 1.5656 ± 0.0163 | 2.2832 ± 0.0080 | 0.4966 ± 0.0011 | 0.6491 ± 0.0040 | 0.6333 ± 0.0013 |

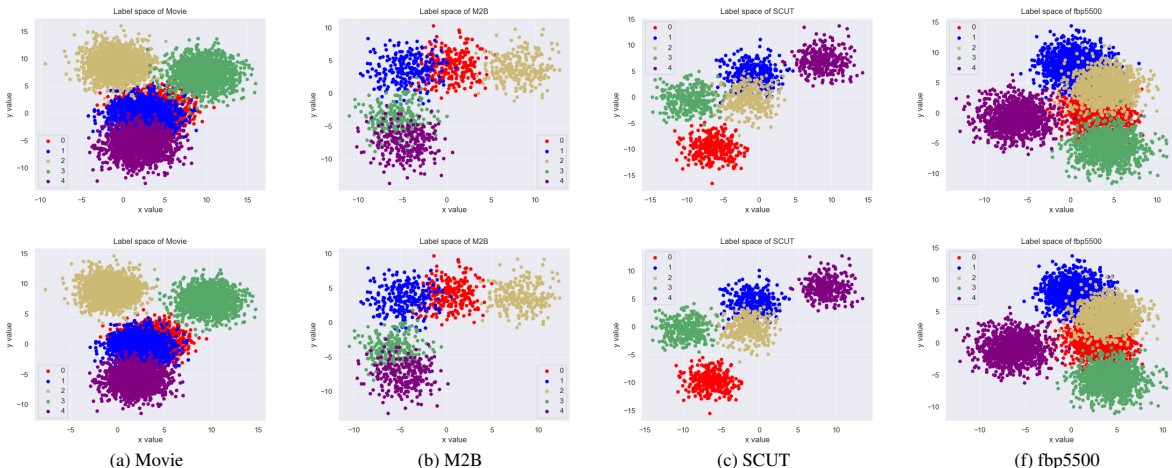

|  (a) Movie | (b) M2B | (c) SCUT | (f) fbp5500 |

Figure 5: This figure visualizes the data distribution in the label space, with the first row indicating the spatial distribution of the raw dataset and the second row indicating the predicted label distribution.

Table 6: The performance of our proposed method with the comparison algorithms on 12 datasets. The best-performing results are marked in **bold**.

| Dataset | Algorithm | Chebyshev ↓ | Clark ↓ | Canberra ↓ | K-L ↓ | Cosine ↑ | Intersection ↑ |
|---------|-----------|-------------|---------|------------|-------|----------|----------------|
| fbp5500 | Ours | **0.1212 ± 0.0001** | **1.1666 ± 0.0123** | **2.0921 ± 0.0200** | **0.1031 ± 0.0012** | **0.9708 ± 0.0012** | **0.8600 ± 0.0035** |
| | Baseline-LDL | 0.1287 ± 0.0091 | 1.1899 ± 0.0333 | 2.1177 ± 0.0432 | 0.1100 ± 0.0033 | 0.9610 ± 0.0022 | 0.8447 ± 0.0064 |
| | INP | 0.1251 ± 0.0002 | 1.1890 ± 0.0120 | 2.0980 ± 0.0223 | 0.1053 ± 0.0009 | 0.9643 ± 0.0015 | 0.8501 ± 0.0025 |
| | PCA | 0.1220 ± 0.0030 | 1.1755 ± 0.0111 | 2.0999 ± 0.0123 | 0.1044 ± 0.0014 | 0.9650 ± 0.0007 | 0.8534 ± 0.0031 |
| | LDL-LRR | 0.1222 ± 0.0030 | 1.1733 ± 0.0038 | 2.0992 ± 0.0095 | 0.1077 ± 0.0077 | 0.9633 ± 0.0021 | 0.8512 ± 0.0066 |
| | LDL-LCLR | 0.1277 ± 0.0016 | 1.1969 ± 0.0039 | 2.1194 ± 0.0046 | 0.1135 ± 0.0006 | 0.9588 ± 0.0044 | 0.8483 ± 0.0014 |
| | LDLSF | 0.1270 ± 0.0028 | 1.1909 ± 0.0164 | 2.1846 ± 0.0119 | 0.1193 ± 0.0041 | 0.9609 ± 0.0019 | 0.8460 ± 0.0007 |
| | LALOT | 0.1306 ± 0.0022 | 1.1921 ± 0.0015 | 2.1111 ± 0.0171 | 0.1120 ± 0.0015 | 0.9430 ± 0.0019 | 0.8400 ± 0.0004 |
| | BFGS-LLD | 0.1299 ± 0.0049 | 1.4655 ± 0.0041 | 2.1675 ± 0.0024 | 0.1135 ± 0.0055 | 0.9595 ± 0.0030 | 0.8419 ± 0.0018 |
| RAF-ML | Ours | **0.1421 ± 0.0025** | **1.3588 ± 0.0321** | **2.6798 ± 0.0026** | **0.2006 ± 0.0033** | **0.9429 ± 0.0023** | **0.8334 ± 0.0029** |
| | Baseline-LDL | 0.1489 ± 0.0023 | 1.3994 ± 0.0451 | 2.7006 ± 0.0903 | 0.2118 ± 0.0022 | 0.9288 ± 0.0019 | 0.8196 ± 0.0044 |
| | INP | 0.1456 ± 0.0021 | 1.3651 ± 0.0441 | 2.6888 ± 0.0023 | 0.2017 ± 0.0012 | 0.9394 ± 0.0026 | 0.8247 ± 0.0077 |
| | PCA | 0.1432 ± 0.0020 | 1.3660 ± 0.0454 | 2.6889 ± 0.0019 | 0.2008 ± 0.0010 | 0.9390 ± 0.0009 | 0.8320 ± 0.0023 |
| | LDL-LRR | 0.1426 ± 0.0033 | 1.3659 ± 0.0211 | 2.7125 ± 0.0422 | 0.2149 ± 0.0007 | 0.9390 ± 0.0013 | 0.8277 ± 0.0044 |
| | LDL-LCLR | 0.1515 ± 0.0022 | 1.5923 ± 0.0117 | 2.7779 ± 0.0239 | 0.2244 ± 0.0030 | 0.9262 ± 0.0062 | 0.8189 ± 0.0098 |
| | LDLSF | 0.1488 ± 0.0024 | 1.3889 ± 0.0086 | 2.7672 ± 0.0660 | 0.2302 ± 0.0044 | 0.9111 ± 0.0051 | 0.8117 ± 0.0022 |
| | LALOT | 0.1479 ± 0.0010 | 1.3659 ± 0.0099 | 2.6956 ± 0.0144 | 0.2221 ± 0.0064 | 0.9311 ± 0.0021 | 0.8107 ± 0.0008 |
| | BFGS-LLD | 0.1499 ± 0.0009 | 1.6656 ± 0.0066 | 2.7101 ± 0.0211 | 0.2541 ± 0.0055 | 0.9204 ± 0.0023 | 0.8157 ± 0.0050 |
| Twitter | Ours | **0.2770 ± 0.0081** | **2.2309 ± 0.0113** | **5.1097 ± 0.0051** | **0.5104 ± 0.0054** | **0.8987 ± 0.0044** | **0.7987 ± 0.0016** |
| | Baseline-LDL | 0.2887 ± 0.0040 | 2.3008 ± 0.0151 | 5.4999 ± 0.1555 | 0.6060 ± 0.0042 | 0.8667 ± 0.0066 | 0.7774 ± 0.0031 |
| | INP | 0.2777 ± 0.0021 | 2.2374 ± 0.0110 | 5.1163 ± 0.0018 | 0.5111 ± 0.0029 | 0.8807 ± 0.0049 | 0.7891 ± 0.0014 |
| | PCA | 0.2771 ± 0.0082 | 2.2343 ± 0.0144 | 5.1167 ± 0.0072 | 0.5110 ± 0.0036 | 0.8837 ± 0.0012 | 0.7924 ± 0.0041 |
| | LDL-LRR | 0.2802 ± 0.0021 | 2.2441 ± 0.0051 | 5.2002 ± 0.0023 | 0.5189 ± 0.0035 | 0.8662 ± 0.0042 | 0.7789 ± 0.0014 |
| | LDL-LCLR | 0.2994 ± 0.0045 | 2.4900 ± 0.0012 | 6.9609 ± 0.0041 | 0.6056 ± 0.0031 | 0.7110 ± 0.0021 | 0.7110 ± 0.0088 |
| | LDLSF | 0.3007 ± 0.0002 | 2.7887 ± 0.0057 | 5.6101 ± 0.0118 | 0.6396 ± 0.0022 | 0.7939 ± 0.0098 | 0.7660 ± 0.0007 |
| | LALOT | 0.3133 ± 0.0021 | 2.3141 ± 0.0016 | 5.5336 ± 0.0241 | 0.5233 ± 0.0012 | 0.8595 ± 0.0550 | 0.7214 ± 0.0049 |
| | BFGS-LLD | 0.3114 ± 0.0044 | 2.5511 ± 0.0028 | 5.7145 ± 0.0041 | 0.5461 ± 0.0153 | 0.8335 ± 0.0055 | 0.7744 ± 0.0020 |
| Flickr | Ours | **0.2801 ± 0.0088** | **2.3169 ± 0.0064** | **5.2188 ± 0.0159** | **0.5314 ± 0.0033** | **0.8406 ± 0.0044** | **0.7832 ± 0.0025** |
| | Baseline-LDL | 0.3134 ± 0.0021 | 2.6641 ± 0.1051 | 5.5599 ± 0.0130 | 0.6007 ± 0.0022 | 0.8330± 0.0099 | 0.7661 ± 0.0034 |
| | INP | 0.2816 ± 0.0031 | 2.3356 ± 0.0097 | 5.2222 ± 0.0159 | **0.5314 ± 0.0013** | **0.8406 ± 0.0014** | 0.7741 ± 0.0025 |
| | PCA | 0.2813 ± 0.0074 | 2.3226 ± 0.0061 | 5.2210 ± 0.0103 | 0.5317 ± 0.0072 | 0.8405 ± 0.0055 | 0.7749 ± 0.0092 |
| | LDL-LRR | 0.2885 ± 0.0012 | 2.3209 ± 0.0174 | 5.3459 ± 0.0229 | 0.5558 ± 0.0032 | 0.8401± 0.0040 | 0.7699 ± 0.0037 |
| | LDL-LCLR | 0.2970 ± 0.0009 | 2.4444 ± 0.0063 | 6.1600 ± 0.0041 | 0.6222 ± 0.0013 | 0.7919 ± 0.0029 | 0.7090 ± 0.0070 |
| | LDLSF | 0.3301 ± 0.0009 | 2.8888 ± 0.0459 | 5.9152 ± 0.0121 | 0.6100 ± 0.0021 | 0.8139 ± 0.0098 | 0.7360 ± 0.0037 |
| | LALOT | 0.3411 ± 0.0026 | 2.9140 ± 0.0019 | 5.3333 ± 0.0243 | 0.5737 ± 0.0012 | 0.8225 ± 0.0202 | 0.7144 ± 0.0004 |
| | BFGS-LLD | 0.3200 ± 0.0041 | 2.7517 ± 0.0060 | 5.8149 ± 0.0048 | 0.5961 ± 0.0099 | 0.8131 ± 0.0011 | 0.7407 ± 0.0077 |

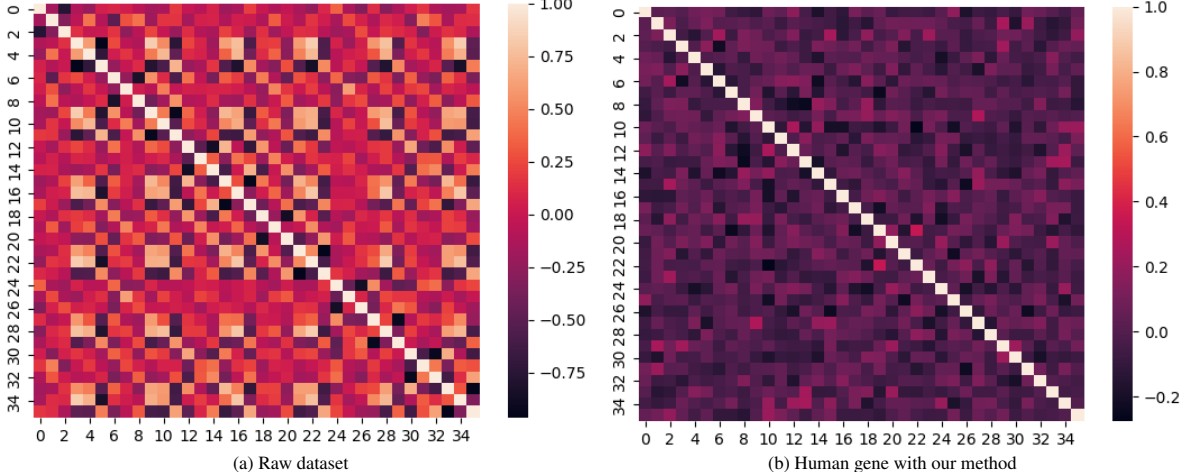

(a) Raw dataset

(b) Human gene with our method

Figure 6: This figure shows the feature correlation of Gene datasets with our method. Our approach has a clear ability to decouple features.

