# OpenReview forum: "Trusted re-weighting for label distribution learning"
_auai.org/UAI/2024/Conference — UAI 2024 poster_

### Official Review · Reviewer_C4ha · 2024-02-27

**Q2-1 Originality-Novelty:** 2
**Q2-2 Correctness-Technical Quality:** 2
**Q2-5 Clarity Of Writing:** 3

**Q1 Summary And Contributions:**

The paper introduces Label Distribution Learning (LDL) as a novel machine learning paradigm for characterizing instances with descriptive degrees rather than binary labels. However, the presence of noise in the label space can degrade the predictive ability of LDL algorithms, especially given the difficulty in accurately annotating each label due to the continuous nature of the descriptive degrees. To tackle this challenge, the proposed stability-trust LDL framework leverages feature decoupling and prototype guidance to reconstruct the feature space of LDL datasets. The framework employs prototype learning to identify reliable cluster centers, filtering out clean samples from the original dataset. The feature space is then decoupled by modeling a weight assigner learned on the clean sample set, eliminating correlations among features and assigning weights to each sample. LDL algorithms can subsequently be trained on this re-weighted dataset, enhancing robust modeling. Experimental results, including validation on a new image dataset, demonstrate the framework's efficacy in boosting the performance of LDL algorithms on datasets with label noise.

**Q2-3 Extent To Which Claims Are Supported By Evidence:**

3: Good: the main claims are supported by convincing evidence (in the form of adequate experimental evaluation, proofs, (pseudo-)code, references, assumptions).

**Q2-4 Reproducibility:**

3: Good: key resources (e.g. proofs, code, data) are available and key details (e.g. proofs, experimental setup) are sufficiently well-described for competent researchers to confidently reproduce the main results.

**Q3 Main Strengths:**

- The research problem is realistic and important.
- The motivation of this paper is clear.
- The method design is overall insightful, following good experimental results.

**Q4 Main Weakness:**

- The theoretical analysis is somewhat weak. The effectiveness of the proposed method is well justified in theory.
- Writing should be polished to improve the legibility of this paper.

**Q5 Detailed Comments To The Authors:**

- The statement of clean samples (with labeled noise) is somewhat confusing. It would be better to state noisy samples. Also, the noisy labels are not equal to incorrect labels. They are a combination of correct and incorrect labels.
- Could the previous works on learning with noisy labels be applied to the research topic of this paper, e.g., [1] and [2]?
- Would label noise influence the prototype learning? If we can use some robust mean/covariance algorithms, will the learning be better or more robust?
- Could the paper provide an algorithm flow to the proposed framework? It will help readers better capture the technical details.
- Could the paper provide more details about the optimization of the objective?
- For Eq (7), which factors bring the constant error?

References
[1] Robust early-learning: Hindering the memorization of noisy labels. ICLR 2021.
[2] Selective-supervised contrastive learning with noisy labels. CVPR 2022.

**Q9 Complying With Reviewing Instructions:**

Yes

---

> ### Author Rebuttal · Authors · 2024-04-04
>
> We sincerely thank the reviewers for their valuable comments.
>
> First, we will fix the description of the noise labels in the manuscript according to the suggestion.
>
> Second, we will cite [1] and [2] in the manuscript to add related work that we have missed.
>
> Third, label noise affects prototype learning, and robust mean-covariance algorithms are considered in our expanded version.
>
> Fourth, we will add more goal optimization details and flowcharts in the revised manuscript.
>
> Finally, for Eq. 7, the ill-condition of the feature matrix (X) introduces a constant error.
>
> [1] Robust early learning: Hindering the memorization of noisy labels. ICLR 2021.
>
> [2] Selective-supervised contrastive learning with noisy labels. CVPR 2022.

---

### Official Review · Reviewer_D9dG · 2024-03-11

**Q2-1 Originality-Novelty:** 2
**Q2-2 Correctness-Technical Quality:** 3
**Q2-5 Clarity Of Writing:** 3

**Q1 Summary And Contributions:**

The authors present a new algorithm to learn the needed classifiers in the "label distribution learning" (LDL) scenario, where instances are described, both in training and testing (inference) time by a probability distribution which reflect their "degree of belonging - descriptive degrees" to the set of possible class-values (labels). While a notable set of algorithms have been proposed to learn classifiers in such scenario since the first paper by 2017, the authors argue that their method is specially suited to deal with the presence of noise in the annotation process in the probabilistic-label space. The proposed algorithm reconstructs the feature space by using decoupling (removing correlation among features) and working with prototypes (instead of full set of samples). In this way, the training set is "recodified-reweighted".

**Q2-3 Extent To Which Claims Are Supported By Evidence:**

3: Good: the main claims are supported by convincing evidence (in the form of adequate experimental evaluation, proofs, (pseudo-)code, references, assumptions).

**Q2-4 Reproducibility:**

3: Good: key resources (e.g. proofs, code, data) are available and key details (e.g. proofs, experimental setup) are sufficiently well-described for competent researchers to confidently reproduce the main results.

**Q3 Main Strengths:**

The proposal of a new algorithm in the relatively new LDL learning scenario.

I appreciate the theoretical analysis of the "feature decoupling" state (removal of correlations among features) of the algorithm (Section 4, last part).

**Q4 Main Weakness:**

I think that the authors do not have a coherent message with the existence of noise in the annotations. While the motivation of the algorithm is that "LDL algorithms may degrade by the presence of noise in the label space" (Abstract), we can also read the following sentence in page 5: "... noisy data are not always noxious and may have positive incentive properties [Li'2022]". I am confused about this, even beigh the "root - motivation" of the proposed algorithm.

The pipeline of the algorithm is composed of a large set of pieces and steps, which make the algorithm complex to follow. I miss a computational time order of this pipeline.

While the authors do an extensive comparison with other LDL methods, I feel that the proposed method is more complex than the rest of compared LDL methods. If this is true, the authors should inform about this. Is the complexity order of compared LDL algorithms similar?

Performed comparison between LDL methods in a set of domains can be assessed by the "multiple comparison statistical setting" of the following paper:
https://jmlr.org/papers/v7/demsar06a.html

I consider that the experimental section can be enriched with the addition of experimentation on synthetic datasets, specially designed, where an "hypothetical" behaviour of the proposed algorithm is expected.

While the authors defend in the first sentence of the abstract that "Label Distribution Learning (LDL) is a novel machine learning paradigm...", I do not agree with this sentence. While the first paper presenting this learning scenario and an algorithm to learn a classifier over it dates from 2017, a considerable set of methodological proposals can be proposed since 2017: just consult the set of references and compared algorithms. The learning scenario is already "populated by proposals" and, due to the complexity and large number of steps and decisions of the proposed pipeline, I give a lower value to the proposed algorithm.

**Q5 Detailed Comments To The Authors:**

Please check previous Q3 and Q4 sections.

**Q9 Complying With Reviewing Instructions:**

Yes

---

> ### Author Rebuttal · Authors · 2024-04-04
>
> We sincerely thank the reviewers for their valuable comments.
>
> ## Motivation.
>
> A large body of paper demonstrates that noise can be harmful or beneficial, depending on how it is used. In this paper, we hypothesize that label noise refers to the wrong label, which is opposite to the truth. For example, a label distribution is [0.1, 0.2, 0.7] and its noisy label is [0.7, 0.2, 0.1]. In contrast, the noise that we utilize to enhance the generalization ability of the model is non-contradictory. For example, a label distribution is [0.1, 0.2, 0.7] and its positive incentive noise is [0.12, 0.21, 0.67]. Here, the positive incentive noise is similar to the role of label smoothing. In summary, I think our root motives are correct, and it is only possible that our expression has created an ambiguity.
>
> ## Computational time.
>
> As a suggestion, we count the running time of our pipelines. Most of the computational resources are performed on the RTX 3090 GPU shader. We use CUDA acceleration on the PyTorch 1.7 framework.
> | Method   | Data Clearing (ms) | Sample Weighting (h) |
> | -------- | ------------------ | -------------------- |
> | Wc-LDL   | 0.021              | 2                    |
> | JAFFE   | 0.012              | 1                    |
> | SBU-3DFE | 0.050              | 5                    |
> | Scene    | 0.032              | 3                    |
> | Gene     | 0.110              | 8                    |
> | Movie    | 0.060              | 6                    |
> | M2B      | 0.032              | 3                    |
> | SCUT     | 0.047              | 4                    |
> | Fbp5500  | 0.025              | 2                    |
> | RAF-ML   | 0.022              | 2                    |
> | Twitter  | 0.049              | 4                    |
> | Flickr   | 0.051              | 5                    |
> Although our preprocessing method requires some computational time, quality features can help downstream LDL algorithms save training time to reach SOTA. For example, LDL-LRR takes more than 12 hours to train, but with the help of our reconstructed features, its training time can be saved by 3 hours to reach SOTA.
>
>
> ## Multiple comparison statistical setting.
>
> We thank the reviewers for providing Statistical Comparisons of Classifiers over Multiple Data Sets to help us analyze the algorithm performance and robustness in depth. In the revised manuscript, we will add the results of the Wilcoxon signed ranks test, the Friedman test, and other test methods.
>
> ## The experimental section can be enriched.
>
> As a suggestion, we experiment on the wc-LDL dataset by conducting different intensities of noise to the labels. Specifically, for each distribution value, we add a Gaussian noise whose mean is the value of the label distribution with variances of 0.1,0.3, and 0.5, respectively.
>
> | Gaussian Noise Variance | Ours            | Baseline-LDL    | INP             |
> | ----------------------- | --------------- | --------------- | --------------- |
> | 0                       | 0.0743 ± 0.0011 | 0.0788 ± 0.0019 | 0.0779 ± 0.0021 |
> |                         | 0.3884 ± 0.0055 | 0.4008 ± 0.0042 | 0.3980 ± 0.0051 |
> |                         | 0.7667 ± 0.0033 | 0.7770 ± 0.0023 | 0.7779 ± 0.0030 |
> |                         | 0.0421 ± 0.0008 | 0.0408 ± 0.0056 | 0.0404 ± 0.0020 |
> |                         | 0.9896 ± 0.0009 | 0.9801 ± 0.0017 | 0.9883 ± 0.0009 |
> |                         | 0.8813 ± 0.0014 | 0.8760 ± 0.0015 | 0.8778 ± 0.0014 |
> | 0.1                     | 0.0761 ± 0.0033 | 0.0792 ± 0.0223 | 0.0782 ± 0.0023 |
> |                         | 0.3887 ± 0.0059 | 0.4011 ± 0.0056 | 0.3991 ± 0.0099 |
> |                         | 0.7673 ± 0.0021 | 0.7779 ± 0.0033 | 0.7783 ± 0.0008 |
> |                         | 0.0435 ± 0.0019 | 0.0410 ± 0.0067 | 0.0422 ± 0.0011 |
> |                         | 0.9890 ± 0.0009 | 0.9789 ± 0.0016 | 0.9870 ± 0.0073 |
> |                         | 0.8725 ± 0.0123 | 0.8750 ± 0.0173 | 0.8769 ± 0.0079 |
> | 0.3                     | 0.0799 ± 0.0025 | 0.0812 ± 0.0252 | 0.0805 ± 0.0012 |
> |                         | 0.3921 ± 0.0040 | 0.4210 ± 0.0055 | 0.4022 ± 0.0081 |
> |                         | 0.7693 ± 0.0055 | 0.7799 ± 0.0034 | 0.7881 ± 0.0145 |
> |                         | 0.0442 ± 0.0032 | 0.0442 ± 0.0033 | 0.0445 ± 0.0022 |
> |                         | 0.9754 ± 0.0012 | 0.9650 ± 0.0053 | 0.9620 ± 0.0033 |
> |                         | 0.8611 ± 0.0231 | 0.8522 ± 0.0190 | 0.8620 ± 0.0101 |
> | 0.5                     | 0.0822 ± 0.0020 | 0.0889 ± 0.0270 | 0.0825 ± 0.0051 |
> |                         | 0.4030 ± 0.0044 | 0.4401 ± 0.0150 | 0.4523 ± 0.0088 |
> |                         | 0.7890 ± 0.0056 | 0.7900 ± 0.0031 | 0.8020 ± 0.0149 |
> |                         | 0.0459 ± 0.0122 | 0.0452 ± 0.0049 | 0.0459 ± 0.0072 |
> |                         | 0.9600 ± 0.0033 | 0.9410 ± 0.0155 | 0.9421 ± 0.0063 |
> |                         | 0.8511 ± 0.0220 | 0.8319 ± 0.0159 | 0.8325 ± 0.0171 |

---

### Official Review · Reviewer_Le7L · 2024-03-23

**Q2-1 Originality-Novelty:** 3
**Q2-2 Correctness-Technical Quality:** 3
**Q2-5 Clarity Of Writing:** 3

**Q1 Summary And Contributions:**

The paper introduces a novel framework for LDL aimed at addressing the challenges posed by noise in label space. Label distribution learning (LDL) shifts from traditional 0/1 labels to descriptive degrees between 0 and 1, representing the degree to which each label describes an instance. However, accurately annotating these descriptive degrees is challenging, leading to noise in the label space which degrades the performance of LDL algorithms. To tackle this, the authors propose a stability-trust LDL framework that reconstructs the feature space of LDL datasets using feature decoupling and prototype guidance. This framework selects reliable cluster centers as representative vectors of label distributions, filters out clean samples, and decouples the feature space by assigning weights to samples, thus enhancing robust modeling of LDL algorithms. Experimental results on a new image dataset demonstrate the effectiveness of the proposed framework in improving LDL algorithm performance in the presence of label noise.

**Q2-3 Extent To Which Claims Are Supported By Evidence:**

3: Good: the main claims are supported by convincing evidence (in the form of adequate experimental evaluation, proofs, (pseudo-)code, references, assumptions).

**Q2-4 Reproducibility:**

2: Fair: key resources (e.g. proofs, code, data) are unavailable but key details (e.g. proof sketches, experimental setup) are sufficiently well-described for an expert to confidently reproduce the main results.

**Q3 Main Strengths:**

- The paper presents a novel combination of prototype learning and feature decoupling to tackle the challenge of label noise in LDL, marking a significant advancement in the field. Unlike existing methods that embed noise processing within the prediction algorithm, this framework offers a generalized pre-processing approach that can enhance the performance of various existing LDL algorithms.
- The proposed framework efficiently filters out clean samples and decouples feature space, facilitating faster convergence and robust model performance without requiring extensive knowledge or sampling processes.
- The paper conducts an extensive evaluation of the proposed method across several datasets, including a newly created image dataset, showcasing its applicability and effectiveness in different contexts.

**Q4 Main Weakness:**

- The method involves multiple steps, including prototype learning and feature decoupling, which might complicate the implementation process. The effectiveness of the framework heavily relies on the accurate selection of prototypes, which could vary significantly across different datasets.
- The paper could benefit from a more detailed analysis of the framework's performance across varying levels of label noise, providing deeper insights into its robustness and limitations.

**Q5 Detailed Comments To The Authors:**

N/A.

**Q9 Complying With Reviewing Instructions:**

Yes

---

> ### Author Rebuttal · Authors · 2024-04-04
>
> We sincerely thank the reviewers for their valuable comments. As a suggestion, we experiment on the wc-LDL and JAFFE datasets by conducting different intensities of noise to the labels. Specifically, for each distribution value, we add a Gaussian noise whose mean is the value of the label distribution with variances of 0.1,0.3, and 0.5, respectively.
>
> The results of the experiment on dataset Wc-LDL (Chebyshev ↓ Clark ↓ Canberra ↓ K-L ↓ Cosine ↑ Intersection ↑) are as follows:
>
> | Gaussian Noise Variance | Ours            | Baseline-LDL    | INP             |
> | ----------------------- | --------------- | --------------- | --------------- |
> | 0                       | 0.0743 ± 0.0011 | 0.0788 ± 0.0019 | 0.0779 ± 0.0021 |
> |                         | 0.3884 ± 0.0055 | 0.4008 ± 0.0042 | 0.3980 ± 0.0051 |
> |                         | 0.7667 ± 0.0033 | 0.7770 ± 0.0023 | 0.7779 ± 0.0030 |
> |                         | 0.0421 ± 0.0008 | 0.0408 ± 0.0056 | 0.0404 ± 0.0020 |
> |                         | 0.9896 ± 0.0009 | 0.9801 ± 0.0017 | 0.9883 ± 0.0009 |
> |                         | 0.8813 ± 0.0014 | 0.8760 ± 0.0015 | 0.8778 ± 0.0014 |
> | 0.1                     | 0.0761 ± 0.0033 | 0.0792 ± 0.0223 | 0.0782 ± 0.0023 |
> |                         | 0.3887 ± 0.0059 | 0.4011 ± 0.0056 | 0.3991 ± 0.0099 |
> |                         | 0.7673 ± 0.0021 | 0.7779 ± 0.0033 | 0.7783 ± 0.0008 |
> |                         | 0.0435 ± 0.0019 | 0.0410 ± 0.0067 | 0.0422 ± 0.0011 |
> |                         | 0.9890 ± 0.0009 | 0.9789 ± 0.0016 | 0.9870 ± 0.0073 |
> |                         | 0.8725 ± 0.0123 | 0.8750 ± 0.0173 | 0.8769 ± 0.0079 |
> | 0.3                     | 0.0799 ± 0.0025 | 0.0812 ± 0.0252 | 0.0805 ± 0.0012 |
> |                         | 0.3921 ± 0.0040 | 0.4210 ± 0.0055 | 0.4022 ± 0.0081 |
> |                         | 0.7693 ± 0.0055 | 0.7799 ± 0.0034 | 0.7881 ± 0.0145 |
> |                         | 0.0442 ± 0.0032 | 0.0442 ± 0.0033 | 0.0445 ± 0.0022 |
> |                         | 0.9754 ± 0.0012 | 0.9650 ± 0.0053 | 0.9620 ± 0.0033 |
> |                         | 0.8611 ± 0.0231 | 0.8522 ± 0.0190 | 0.8620 ± 0.0101 |
> | 0.5                     | 0.0822 ± 0.0020 | 0.0889 ± 0.0270 | 0.0825 ± 0.0051 |
> |                         | 0.4030 ± 0.0044 | 0.4401 ± 0.0150 | 0.4523 ± 0.0088 |
> |                         | 0.7890 ± 0.0056 | 0.7900 ± 0.0031 | 0.8020 ± 0.0149 |
> |                         | 0.0459 ± 0.0122 | 0.0452 ± 0.0049 | 0.0459 ± 0.0072 |
> |                         | 0.9600 ± 0.0033 | 0.9410 ± 0.0155 | 0.9421 ± 0.0063 |
> |                         | 0.8511 ± 0.0220 | 0.8319 ± 0.0159 | 0.8325 ± 0.0171 |
>
> In the same way, all types of algorithms show a decline in performance on the JAFFE dataset, while our method remains SOTA. In addition, we found an interesting phenomenon as the noise level increases the error bar of all types of algorithms is getting bigger overall. In the revised manuscript, we will add and discuss the experiments on noise robustness.

---

### Meta-Review · Area_Chair_ddzp · 2024-04-13

This paper introduces a novel algorithm for label distribution learning, particularly for scenarios with potential label noise, which is an interesting and important problem with many practical applications. The authors propose a re-weighting-based method to address this problem. All the reviewers unanimously agree that this work is of high quality, but nevertheless also raised some suggestions on the paper presentation and several technical discussions. The authors are also encouraged to take reviewers' feedback into account to enhance the clarity of the paper further.